# Microglia NLRP3 Inflammasome and Neuroimmune Signaling in Substance Use Disorders

**DOI:** 10.3390/biom13060922

**Published:** 2023-05-31

**Authors:** Ming-Lei Guo, Soheil Kazemi Roodsari, Yan Cheng, Rachael Elizabeth Dempsey, Wenhui Hu

**Affiliations:** 1Drug Addiction Laboratory, Department of Pathology and Anatomy, Eastern Virginia Medical School, Norfolk, VA 23507, USA; roodsask@evms.edu (S.K.R.); chengy@evms.edu (Y.C.); dempsere@evms.edu (R.E.D.); 2Center for Integrative Neuroscience and Inflammatory Diseases, Eastern Virginia Medical School, Norfolk, VA 23507, USA; 3Center for Metabolic Disease Research, Department of Pathology and Laboratory Medicine, Lewis Katz School of Medicine, Temple University, Philadelphia, PA 19140, USA; wenhui.hu@temple.edu

**Keywords:** inflammasome, NLRP3, abused drugs, cocaine, morphine, neuroinflammation, microglia

## Abstract

During the last decade, substance use disorders (SUDs) have been increasingly recognized as neuroinflammation-related brain diseases. Various types of abused drugs (cocaine, methamphetamine, alcohol, opiate-like drugs, marijuana, etc.) can modulate the activation status of microglia and neuroinflammation levels which are involved in the pathogenesis of SUDs. Several neuroimmune signaling pathways, including TLR/NF-кB, reactive oxygen species, mitochondria dysfunction, as well as autophagy defection, etc., have been implicated in promoting SUDs. Recently, inflammasome-mediated signaling has been identified as playing critical roles in the microglia activation induced by abused drugs. Among the family of inflammasomes, NOD-, LRR-, and pyrin-domain-containing protein 3 (NLRP3) serves the primary research target due to its abundant expression in microglia. NLRP3 has the capability of integrating multiple external and internal inputs and coordinately determining the intensity of microglia activation under various pathological conditions. Here, we summarize the effects of abused drugs on NLRP3 inflammasomes, as well as others, if any. The research on this topic is still at an infant stage; however, the readily available findings suggest that NLRP3 inflammasome could be a common downstream effector stimulated by various types of abused drugs and play critical roles in determining abused-drug-mediated biological effects through enhancing glia–neuron communications. NLRP3 inflammasome might serve as a novel target for ameliorating the development of SUDs.

## 1. Introduction

Microglia, the brain-resident macrophages, generally account for 5–12% of all brain cells, with varying density in the different brain regions of rodents [1]. Previous studies showed that microglia constitute 5% in the cerebral cortex and in the corpus callosum, and around 12% in the substantia nigra of the mouse brain [1]. In the human brain, the variability in microglia density in different regions is even wider, with about 0.3% in the gray matter of the cerebellum and 11% in the medulla oblongata [2], and about 5% in cortical gray matter [3]. Microglia constitute the critical component in the first-line-of-defense system and perform constitutional immune surveillance in the central nervous system (CNS) [4,5]. Under physical conditions, microglia play essential roles in regulating brain development, as well as maintaining the homeostasis of the adult brain through interacting with neurons, astrocytes, and oligodendrocytes [6,7]. Meanwhile, microglia are sensitive to various types of stimuli and can be quickly changed to activation status. Multiple pro- and anti-inflammatory neuroimmune signaling pathways have been demonstrated to coordinately regulate the status of microglia activation [6,7]. Abnormal microglia activation (neuroinflammation) has been implicated as a major risk factor contributing to the pathogenesis of multiple neurodegenerative diseases including Alzheimer’s diseases (ADs) [8,9], Parkinson’s diseases (PDs) [10,11], amyotrophic lateral sclerosis (ALS) [12], as well as recently SUDs [13,14]. NLRP3 inflammasome belongs to the superfamily of pattern-recognition receptors (PRRs) recognizing pathogen-associated molecular patterns. The unique feature of NLRP3 activation is its two-step process: priming and inflammasome assembly. NLRP3 could serve as a hub integrating multiple signals to determine the intensity of microglia activation [15,16]. Accumulating evidence shows that abused drugs, including cocaine, methamphetamine (Meth), alcohol, opiate-like drugs, and marijuana, are capable of interacting with NLRP3 inflammasome through either signal 1 or signal 2 pathways. The understanding of microglia biology, inflammasome signaling, and the involvement of microglia in SUDs has advanced substantially during the last decade. Here, we summarize these advances with a focus on the effects of abused drugs on NLRP3 inflammasomes from both in vitro and in vivo studies. The available findings suggest that NLRP3 inflammasome might be the common downstream effector of most abused drugs, if not all, and targeting NLRP3 inflammasomes might provide a novel therapeutic approach for SUDs.

## 2. Microglia, Inflammasomes, and SUDs

### 2.1. Updates on Microglia Biology

After the first discovery of microglia in 1919, there had been not much progress on microglia biology in the following sixty years due to technical limitations. However, in the last twenty years, the basic understanding of microglia has been significantly advanced for their functions and heterogeneity in vivo [4]. Microglia are traditionally believed to be immunocompetent cells and to maintain quiescent state under basal conditions. Microglia are sensitive to various types of internal and external stimuli. Upon stimulation, microglia quickly adopt activation status and produce and secrete a plethora of cytokines and chemokines leading to increased neuroinflammation levels [4,5]. Microglia are also crucial for maintaining the normal function of neurons. In the development stage, microglia actively interact with neurons for synapse pruning (synapse elimination) to ensure proper neuroplasticity and brain development. In adult brains, even at basal levels, microglia are still very active in patrolling around and surveying microenvironments through their long and thin processes. Basically, microglia function in multiple roles as housekeepers, guards, and warriors to maintain brain homeostasis and ensure normal brain functions [7,17]. The understanding of microglia activation status has also been greatly revised. Microglia were previously assumed to fall into three different functional statuses: M0 (inactive), M1 (pro-inflammatory), and M2 (anti-inflammatory); however, such a classification is too simple or arbitrary to explain the roles of microglia in physiological or pathological conditions. Currently, microglia are believed to exist more in a continuum of states from pro-inflammatory to anti-inflammatory status with many intermediate states. Based on the presence/absence of stimulation, microglia can be grouped into at least four functional statuses based on their gene-expression profile and morphological changes: quiescence, priming, partial activation, and full activation. As for their heterogeneity, microglia are now well-recognized as having differences throughout the brain. The numbers, sizes, morphology, and immune responses of microglia have substantial differences based on their brain location [18,19]. For example, the degradation ability of microglia in the cortex and cerebellum is different due to their different lysosome functions [20,21]. In the past five years, microglia have been identified as belonging to novel subsets based on their transcriptional profiles (single-cell RNA sequencing) under various physiological/pathological conditions. A novel subset called disease-associated microglia (DAM) or microglial neurodegenerative phenotype (MGnD) has been identified in the brains of mouse models with Alzheimer’s diseases and Parkinson’s diseases [22,23]. In addition, proliferation-associated microglia (PAM), neurodegeneration-associated microglia, lipid-droplet-accumulation microglia, etc., have been identified in various disease models [24,25,26]. Such subsets play critical roles (either bad or good) in the pathogenesis of various types of brain diseases [24,25,26]. The main discoveries about microglia during the past hundred years have been summarized in Figure 1.

### 2.2. NLRP3 Inflammasome Pathway

Multiple neuroimmune signaling pathways have been shown to participate in microglia activation. Among them, the CX3CR1/CX3CL1 axis, CD200/CD200R, TGFβ, NF-кB pathway, toll-like receptors (TLRs), and inflammasome signaling have been well-investigated and shown to restrain or promote microglia activation, and they have been well-reviewed elsewhere [27,28,29]. These pro- and anti-inflammatory signaling pathways mutually interact and determine the intensity of microglia activation in a concerted manner.

The superfamily of inflammasomes, particularly NLRP3 inflammasome, have been occupying the central stage for research on inflammation-related diseases in the past decade [15,16]. Briefly, NLRP3 inflammasome activation needs two different signals: signal 1 is for increasing the expression of NLRP3 as well as proIL18 and proIL1β (priming). The most well-known signal 1 is the TLR/NF-кB pathway. Signal 2 is for the assembling of NLRP3 inflammasome, which includes NLRP3, ASC (apoptosis-associated speck-like protein), and procaspase-1 (pCasp1). The whole complex together leads to the self-cleavage of pCasp1 to release mature caspase 1 (mCasp1). Then, mCasp 1 processes pro-IL1β and pro-IL18 into mIL1β and mIL18, respectively. Numerous signals have been identified as signal 2, including reactive oxygen species (ROS), K^+^ influx, P2Y receptors, mitochondrial defection, and lysosomal disruption, etc., to increase the formation of NLRP3 inflammasome [30]. The mCasp1 also cleaves gasdermin d (GADMD) to form GADMD pores in the membrane, allowing the release of mIL18 and mIL1β (pyroptosis). This is called the NLRP3 canonical pathway for NLRP3. There is also a non-canonical pathway: lipopolysaccharide (LPS) activates caspase 11, leading to the formation of NLRP3 inflammasome. A schematic of signal 1 and signal 2 and the canonical and non-canonical pathways of NLRP3 is shown in Figure 2.

### 2.3. SUDs and Neuroinflammation

SUDs have been traditionally believed to be a neuroplastic brain disorder, and great effort has been put into exploring the mechanisms responsible for the changes in neuronal plasticity as well as brain circuitry during the pathogenesis of SUDs in the past three decades. However, the neuron-centered hypothesis has not produced any breakthroughs in the treatment of SUDs, and no FDA-approved drugs are available to block the development of SUDs, especially for cocaine-use disorders. This dilemma has resulted in a hypothesis that other types of brain cells, such as glial cells and glia–neuron communications, could contribute equally to SUDs. Recently, SUDs have been increasingly appreciated to be neuroinflammation-related brain disorders. Increased microglia activation has been identified in the brains of rodents exposed to multiple types of abused drugs [31,32,33]. Human studies also showed aberrant expression profiles for cytokines in the serum, as well as microglia dysregulation in the postmortem brains, of addicts. For example, increased microglia activation, increased IL6, and decreased IL10 levels in serum were found in human addicts [34]. Increased levels of brain-derived neurotrophic factor, IL1β, and tumor necrosis factor α (TNFα) were revealed in serum obtained from cocaine addicts [35]. Microglia inhibition is capable of blocking abused-drugs-mediated behavioral changes relevant to reward effects. Furthermore, targeting microglia and modulating the strength of neuroimmune signaling have been suggested as novel therapeutic approaches for SUDs [36,37].

## 3. The Effects of Abused Drugs on Microglia and Inflammasomes

### 3.1. The Effects of Cocaine on Neuroimmune Signaling and NLRP3 Inflammasome

Cocaine is a potent psychostimulant and one of the most abused drugs in the United States. According to the 2021 National Survey on Drug Use and Health, it has been estimated that in the past 12 months, 5.2 million individuals in the USA aged 12 and older have used cocaine, and approximately 20,000 people have died from a cocaine-related overdose (https://nida.nih.gov/publications/research-reports/cocaine/what-scope-cocaine-use-in-united-states, access on 24 April 2023).

In addition to the effects on the dopamine system in the brain, cocaine is known to dysregulate inflammation levels in both the CNS and peripheral systems. In a chronic cocaine abuser, there is a significant increase in IL6 and decrease in IL10 levels in the serum, indicating peripheral inflammation [34]. In the CNS, multiple pathways have been identified as being responsible for cocaine-mediated microglia activation. Cocaine is capable of increasing the expression of toll-like receptor-2 (TLR2) and ROS levels in BV2 cells [38]. Cocaine upregulates endoplasmic reticulum (ER) stress, evidenced by the increased expression levels of phosphorylated protein kinase R-like endoplasmic reticulum kinase (pPERK), phosphorylated eukaryotic initiation factor 2α (peIF2α), and activating transcription factor 4 (ATF4) [38]. TLR4 and its downstream signaling NF-кB are implicated in cocaine-mediated microglia activation [39,40]. Besides these classical neuroimmune signaling pathways, microRNA (miRNA) dysregulation has been implicated as another mechanism responsible for microglia activation induced by cocaine. Mir-124, the most abundant brain miRNA, is decreased in microglia with cocaine exposure, and overexpression of miR-124 mitigates cocaine-mediated TLR4 signaling strength, resulting in microglia inhibition [41]. Mechanically, an increased level of miRNA-124 promoter methylation is responsible for cocaine-mediated downregulation of miR-124 [42]. Autophagy dysregulation is also involved in cocaine-mediated microglia activation. Cocaine increases the expression levels of autophagy-related proteins, including beclin1, ATG5, and LC3II, and autophagy inhibition could partially block cocaine-mediated microglia activation [43].

The TLRs/NF-кB axis, ROS, and autophagy defection could contribute to NLRP3 inflammasome activation, which implies that cocaine has the ability to modulate NLRP3 inflammasome activity [44]. Indeed, emerging evidence suggests that cocaine could upregulate NLRP3 inflammasome activity. In human macrophages, cocaine increased NLRP3 levels, and cocaine and HIV infection exerted synergistic upregulation effects on the levels of NLRP3 and IL1β [45]. Cocaine also upregulated NLRP3 inflammasome activity in microglia (BV2 cells and mouse primary microglia), and both a genetic approach (siRNA NLRP3) and a pharmacological approach (MCC950) mitigated cocaine-mediated microglia activation [46]. Mechanically, increased ROS production and sigma 1 receptor seemed to be involved in cocaine-mediated upregulation of NLRP3 [46]. The upregulation of NLRP3 inflammasome activity seems also important in cocaine-mediated reward effects. CX3CR1-deficiency mice showed greater enhancement in cocaine-mediated hyperlocomotion and conditional place preference than WT mice did [47]. The CX3CR1/CX3CL1 axis maintains microglia in quiescence and CX3CL1 was capable of inhibiting NLRP3 inflammasome [48]. Indeed, there was increased NLRP3 inflammasome activity in CX3CR1-deficiency mice compared to WT mice with cocaine administration. These results implied that NLRP3 inflammasome activity is positively associated with cocaine-mediated reward effects [47]. However, such assumptions about the roles of NLRP3 in cocaine addiction need further investigation and verification by using NLRP3-conditional knockout (KO) mice (microglial-specific KO). Unlike NLRP3 inflammasome, the effects of cocaine on the other inflammasomes, including NLRP1, NLRP6, NLRC4, and AIM2, have not been reported till now.

### 3.2. The Effects of Meth on Neuroimmune Signaling and NLRP3 Inflammasome

Meth is another addictive psychostimulant commonly abused. Over 14.7 million people tried Meth at least once between the years of 2015 and 2018, with the death toll rising each year [49].

The effects of Meth on microglia activation and neuroinflammation have been well-addressed [50,51,52,53]. Similar to cocaine, Meth can activate microglia through multiple pathways, including the increased production of ROS/ER stress and the TLRs/MyD88/NF-кB axis. Pellino 1 (Peli1) is highly abundant in microglia and plays critical roles in inducing microglia activation by strengthening the TLRs pathway [54,55]. Recently, the role of the TLR4-TRIF-Peli1 axis has been revealed in Meth-mediated microglia activation [56]. The brain–blood barrier (BBB) is crucial for maintaining brain homeostasis through controlling the crosstalk between the central and peripheral immune systems. Meth is capable of inducing damage to the BBB through decreasing the levels of tight junction proteins ZO-1, occludin, and claudin-5, which ultimately increases neuroinflammation levels [57]. Epigenetic regulation is also involved in Meth-mediated microglia activation [58]. Two miRNAs, miR-142a-3p and miR-155-5p, were found decreased in Meth-exposed microglia, and correspondingly their target Peli1 was increased. Overexpression of these two miRNAs could decrease Peli1 levels and protect Meth-mediated immune responses [58]. In addition, sigma 1 receptor seems also to be involved in this process since its ligand SN79 blocks Meth-mediated microglia activation [59].

NLRP3 inflammasome was also involved in Meth-mediated microglia activation. Meth potentiates the assembly of NLPR3 inflammasome (NLRP3/ASC/pro-caspase protein complex) and increases the production of mIL1β [60]. The blockade of capase-1 activity and lysosomal cathepsin B activity, or inhibition of mitochondrial ROS production, reverse the effects of Meth on immune response and further consolidate the roles of NLRP3 inflammasome in Meth-induced microglia activation [60]. Another investigation showed that NLRP3 inflammasome was implicated in Meth-mediated microglia activation, probably through the miR-143/PUMA axis, although the details of the mechanisms remain very much unclear [61]. Inflammasome upregulation was also found in chronic Meth users. Upregulation of NLRP1 and NLRP3 levels was revealed in the postmortem brain of Meth addicts [62]. Mouse models showed that NLRP3 inhibition could prevent motor deficits and cerebellar degeneration induced by chronic Meth administration, implying the potential therapeutic roles of NLRP3 inflammasome on neurological symptoms in chronic Meth users [63]. NLRP3 inflammasome was suggested to be involved in Meth-mediated intestinal inflammatory injuries [64]. There is a gut–brain axis mediating the crosstalk between gut and brain, and upregulation of intestinal immune responses probably has deleterious effects on neuroinflammation, another route that peripheral NLRP3 inflammasome increases central immune responses [64].

Unlike cocaine, Meth has been reported to have effects on other inflammasomes. NLRP1 inflammasome was involved in Meth-mediated cognitive impairment in rats [65]. However, the effects of Meth on other inflammasomes such as NLRP6, NLRC4, and AIM2 have not been reported till now.

### 3.3. The Effects of Alcohol on Neuroimmune Signaling and NLRP3 Inflammasome

Alcohol is a legally abused substance in most places around the world including the USA. Alcohol abuse/alcohol use disorders (AUDs) have quickly risen as one of the leading causes of death in the United States. According to the 2020 National Survey on Drug Use and Health, about 50% of the population (138.5 million people) aged 12 or older reported drinking alcohol within the past month. Around 22.2% of those individuals (61.6 million people) reported that they had engaged in binge alcohol drinking within the last month and 6.4% (17.7 million people) were heavy alcohol users.

The effects of alcohol on microglia activation and the contributing roles of microglia activation on AUDs have been well-addressed [66,67,68]. Numerous neuroimmune pathways have been revealed in alcohol-induced microglia activation in vitro, including TLR/NF-кB, ROS, high mobility group box 1 (HMICROGLIAB1), etc. Alcohol could also regulate miRNAs, including miR-155, miR-339, and let-7, to modulate microglia activation [69,70,71]. In alcohol-dependent and withdrawal rodents, miR-124 had decreased levels in the limbic forebrain [72]. In addition, alcohol could induce mitochondrial impairment, which further exaggerates neuroinflammation and the subsequent neuronal injuries [73]. More recently, extracellular vehicles (EVs) have been implicated in mediating ethanol-induced inflammatory signaling in microglia [74].

The relationship between alcohol and NLRP3 inflammasome has been well-recognized in the CNS [75,76], and NLRP3 inflammasome inhibition can provide a novel therapeutic approach for the treatment of AUDs [77]. Alcohol is capable of interacting with NLRP3 inflammasome in multiple types of cells including macrophages, peripheral blood mononuclear cells (PBMCs), neurons, and microglia. However, the effects of alcohol on NLRP3 inflammasome seem cellular-context dependent. Alcohol followed with LPS priming could increase the levels of mature IL1β, TNF, and IL6 in human PBMCs [78], whereas in murine macrophage cell line J774, alcohol increased the levels of mature IL1β and IL6 even without LPS priming. Long-term alcohol exposure amplified the release of IL1β upon NLRP3 agonists, but not NLRP1 or AIM2 agonists, indicating the specific effects of alcohol on the NLRP3 inflammasome pathway [78]. The mitochondrial ROS-scavenger MitoQ inhibited the elevated levels of ROS and IL1β induced by chronic alcohol exposure, suggesting that NLRP3 activation is a downstream effector following mitochondrial damage and ROS increase [78]. In neurons, alcohol could act as both signal 1 and signal 2, leading to NLRP3 activation which promotes the pathogenesis of AUDs [79]. Interestingly, alcohol can also induce HMICROGLIAB1 release through NOX2/NLRP1 inflammasome in neurons [80]. In microglia, chronic alcohol treatment enhances TLR4-mediated activation of NLRP3 inflammasome, which is involved in leucocyte infiltration through the brain–blood barrier [81].

In addition to the CNS, the interactions between alcohol and inflammasomes are also evident in peripheral organs including the liver. Alcohol-mediated liver diseases are involved in various types of inflammasome including NLRP3, NLRP6, and NLRC4 inflammasomes [82]. NLRP3 inflammasome plays critical roles in alcohol-mediated steatohepatitis [83]. Intriguingly, NLRP6 inflammasome plays protective roles in alcohol-induced liver diseases. NLRP6 knockout mice show lesser degrees of alcohol-induced liver diseases. However, the direct effects of alcohol on NLRP6 inflammasome have not been revealed [84,85]. In the liver, alcohol might also activate NLRC4 inflammasome since Nlrc4(-/-) mice had greatly reduced activation of IL1β [86].

### 3.4. The Effects of Marijuana on Neuroimmune Signaling and NLRP3 Inflammasome

Marijuana, or cannabis, is the most commonly used illicit recreational drug in North America with the movement towards decriminalization and legalization [87]. According to NIDA in 2020, 17.9% (49.6 million people) of the population aged 12 and older reported using cannabis and 5.1% (14.2 million people) had a cannabis-use disorder (CUD). NIDA estimated that in 2021, 7.1% children in the 8th grade had access to and used cannabis in the past year.

Cannabis herb contains the psychoactive constituent Δ-9 tetrahydrocannabinol (THC), which was historically classified as a hallucinogen [88]. In addition, cannabis plant contains cannabidiol (CBD) which is considered a non-psychoactive component that attenuates THC behavioral and metabolic effects [89]. THC binds to the GPCR cannabinoid receptors, CB1 and CB2. CB1 is distributed throughout the CNS (cortex, hippocampus, basal ganglia, and cerebellum) and aids in modulating glutamate/GABA release. It also interacts with the dopamine, serotonergic (5-HT), and noradrenergic systems [90,91]. CB2 is expressed by hematopoietic cells and is moderately expressed in specific cortical regions and peripheral cells. CB2 is primarily expressed only when there is active neuroinflammation or microglia activation and has shown potential as a therapeutic target for neurodegenerative diseases [92].

The effects of cannabis on neuroinflammation are mixed. In earlier reports, repeated cannabis exposure could induce the microglia activation underlying cerebellar deficits [93]. However, recent findings reached a consensus that THC may play neuroprotective roles by inhibiting neuroinflammation. Mechanically, THC could mitigate NLRP3 inflammasome activity under stimulus condition, probably through the CB2 receptor [94,95,96]. CBD and THC significantly inhibited NLRP3 inflammasome activation stimulated by LPS and ATP, which in turn reduced levels of IL1β, IL6, IL18, and TNFα in macrophages and HBECs [94]. The CB2 agonist JWH-015 also decreased monocyte IL1β production, similar to THC [97]. THC has similar effects on NLRP3 inflammasome in microglia. BV2 cells being treated with cannabis sativa L. phytocomplex partially attenuated the LPS-induced upregulation of IL1β, IL6, and TNFα [98]. CBD treatment suppressed the secretion of the IL1β and NF-κB signaling pathways in LPS-treated mouse microglia [96,99]. Similarly, activation of the CB2 receptor by the synthetic cannabinoid HU-308 induces autophagy in mouse microglia cells and inhibits NLRP3 activation [100]. In a murine ulcer model, CBD treatment downregulates the expression of molecules associated with the NLRP3 inflammasome pathway [101]. Mechanically, CBD reduces the expression of cytidine/uridine monophosphate kinase 2, which inhibits the formation of oxidized mitochondrial DNA and ultimately suppresses the activation of inflammasomes [101].

Till now, most investigations on the effects of cannabis on inflammasomes have primarily focused on NLRP3 inflammasome. Whether cannabis has effects on other inflammasomes remains very much unknown.

### 3.5. The Effects of Opioids on Neuroimmune Signaling and NLRP3 Inflammasome

Morphine is an opioid drug that is considered to be an effective analgesic for the management of pain in clinic. The percentage of the population using opioid-like drugs has been increasing over the years [102].

The effects of morphine on inflammation are also mixed. Some studies have indicated that morphine is anti-inflammatory through upregulating miR-124 [103,104], while others have demonstrated pro-inflammatory effects on microglia [105,106]. The mechanisms underlying such a discrepancy remain very much unknown but are probably due to different drug regimens, such as factors like exposure time and dose. TLR4-mediated neuroimmune signaling is critical for morphine-mediated neuroinflammation. Morphine was reported to directly bind to TLR4 by docking to the LPS-binding pocket of MD-2 [107]. Following stimulation of the TLR4 pathway, NF-κB is activated and pro-inflammatory cytokines are released [107]. In CNS endothelial cells, morphine activates the TLR4 pathway and, in turn, induces rapid phosphorylation of MAPK p38 and ERK [108]. Mechanically, morphine decreases the ubiquitination of tumor necrosis factor receptor associated factor 6 (a critical mediator of TLR/IL-1 signaling) and phosphorylation of TRAF-activated kinase 1. In BV2 cells, morphine has been shown to increase the production of IL1β and TNFα. Likewise, morphine induces the release of pro-inflammatory cytokines (NO, TNFα, IL1β, and IL6) from the activated mouse microglia via the PKC-Akt-ERK1/2 signaling pathway. In astrocytes, morphine could dysregulate the autophagy process through ER stress-mediated pathways, which in turn leads to astrogliosis and neuroinflammation [109]. In addition, beta-amyloid pathways (β-site cleaving enzyme, amyloid precursor protein, etc.) are also involved in morphine-mediated astrogliosis and neuroinflammation [110].

Morphine has the ability to interact with NLRP3 inflammasome in various types of brain cells. Elevated NLRP3 inflammasome activity was involved in morphine-mediated microglia activation and tolerance [111]. In addition, morphine can directly activate NLRP3 inflammasome, leading to paradoxically prolonged neuropathic pain [112]. Several molecules, including DAMPs, HMICROGLIAB1, and biglycan, and purinergic receptor P2X7R, were involved in morphine-mediated NLRP3 activity and tolerance [113,114]. Another report showed that repeated morphine exposure could increase the expression and phosphorylation of TGFβ activated kinase 1 (TAK1), which leads to an increase in NLRP3 activation [115]. Collaboratively, TLR4 knockout mice demonstrated an attenuated morphine-induced tolerance, inhibited NLRP3 activation, and decreased phosphorylation of TAK1 under chronic morphine administration [115]. Fentanyl, another opiate-like drug, could induce cell-specific activation of NLRP3 inflammasome via TLR4 and opioid receptors in astrocytes and neurons, respectively [116].

## 4. The Potential Therapeutic Effects of NLRP3 Inflammasome in SUDs

Since NLRP3 inflammasome plays critical roles in both peripheral and central inflammation, many small molecules including MCC950 and OLT1177 have been developed for the treatment of inflammation-related diseases by regulating NLRP3 inflammasome activity [117]. The progress of these drugs in clinical trials has been well-reviewed for inflammatory bowel diseases [117] and neurodegenerative diseases including AD, PD, stroke, etc. [118]. SUDs have been increasingly recognized as neuroinflammation-related brain diseases and many small molecules with the capability of reducing neuroinflammation levels have been extensively tested in rodent models. Furthermore, several drugs including minocycline, ibudilast, pioglitazone, N-acetylcysteine, and pentoxifylline have advanced in clinical trials for SUDs treatment to different stages [37]. A summary of the effects of these molecules on SUDs in clinical trials and their mechanisms of action is found in Table 1. Interestingly, these molecules also have the capability of modulating NLRP3 inflammasome in various disease models [119,120,121,122,123]. Surprisingly, none of the known NLRP3 inflammasome inhibitors have been tested in SUDs in either pre-clinical or clinical tests. It would be worth exploring the effects of those NLRP3 inhibitors on SUD development.

## 5. Conclusions

Abused drugs can activate microglia through multiple neuroimmune signaling pathways including NLRP3 inflammasome. NLRP3 inflammasome might function as a common downstream effector activated by various types of abused drugs and play critical roles in the pathogenesis of SUDs. The effects of abused drugs on NLRP3 signal 1 and signal 2 pathways have been summarized in Table 2. Targeting NLRP3 inflammasome might provide a novel therapeutic approach for ameliorating the neurological symptoms of SUDs. More investigations should be carried out to test the effects of those NLRP3 inhibitors on SUDs. Currently, there is no direct and consolidated in vivo data to demonstrate that microglia NLRP3 could promote SUD pathogenesis. In addition to NLRP3, microglia express several other inflammasomes and the involvement of other inflammasomes in SUDs remains very much unexplored. Furthermore, the NLRP3 inflammasome is expressed in other types of brain cells, including astrocytes and neurons [124,125]. Thus, to answer the questions about the specific effects of microglia NLRP3 on SUDs, the generation of microglial-specific NLRP3 knockout mice is an emergent need for future investigations.

## Figures and Tables

**Figure 1 biomolecules-13-00922-f001:**
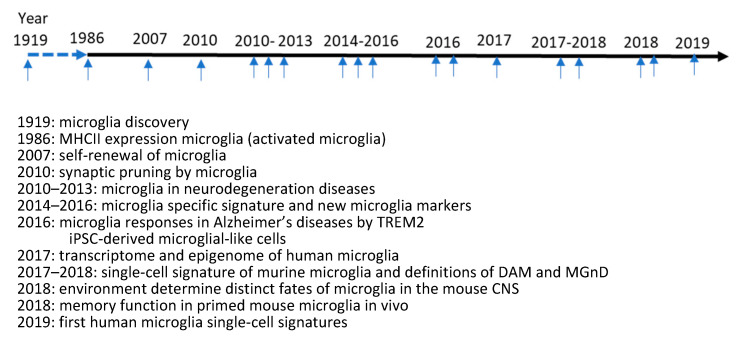
The main discoveries about microglia biology during the last one hundred years.

**Figure 2 biomolecules-13-00922-f002:**
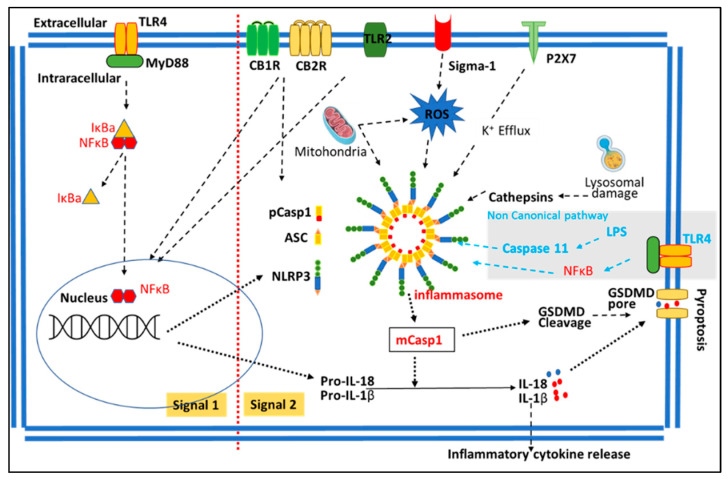
A schematic diagram of signal 1 and signal 2 for NLRP3 inflammasome activation. The grey area is the NLRP3 non-canonical pathway.

**Table 1 biomolecules-13-00922-t001:** The effects of molecules on SUDs in clinical trials as well as NLRP3 inhibition (down arrow: downregulation; N/A: no effects).

Drugs	Mechanisms of Action	Opioids	Psychostimulants	Alcohol	Cannabis	NLRP3 Inhibition
Minocycline	microglial inhibitor	positive effects 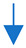	positive effects 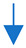	positive effects (-)	N/A	yes
		craving (-)				
Ibudilast	TNFα inhibitor	positive effects (-)	N/A	positive effects (-)	N/A	yes
		withdrawal 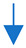		craving 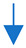		
Pioglitazone	cytokine inhibitor	positive effects (-)	reinforcing effects 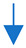	N/A	N/A	yes
		craving 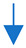	craving 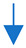			
N-Acetylcysteine	GLT-1 upregulation	N/A	positive effects 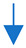	N/A	craving 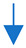	yes
	ROS scavenger		craving 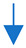		abstinence (-)	
			abstinence 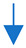			
Pentoxifylline	cytokine inhibitor	N/A	abstinence (-)	N/A	N/A	yes

**Table 2 biomolecules-13-00922-t002:** Summary of the effects of abused drugs on NLRP3 signaling and miR-124.

	Cocaine	Meth	Alcohol	Marijuana	Morphine
**TLR/NF-кB**	Up	Up	Up	Down	Up
**ROS**	Up	Up	Up	No test	Up
**NLRP3**	Up	Up	Up	Down	Up
**miR-124**	Down	Down	Down	No test	Up

## Data Availability

Not applicable.

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
