# Peer review of "Microglia NLRP3 Inflammasome and Neuroimmune Signaling in Substance Use Disorders"

_biomolecules, 2023, doi:10.3390/biom13060922_

Round 1
Reviewer 1 Report
The review manuscript by Guo et al. with a title “Microglia NLRP3 Inflammasome Activation and Neuroimmune Signaling in Substance Use Disorders” is a comprehensive review of the recent literature on drug abuse-induced neuroimflammation. In particular, the review focuses on the signaling pathway of activated microglia in the brain and drugs such as cocaine, methamphetamine, alcohol, opiate-like drugs and marijuana. The review begins by explaining the role of microglia and NLRP3 inflammasome signaling in neuroinflammation. This is followed by several sections, each dedicated to one drug. The review is well and clearly structured with each paragraph having a similar structure. As not many studies have been conducted evaluating the role of abused drugs in neuroinflammation, this interesting review may inspire researchers to conduct further studies in this area of research. The manuscript can be published after some minor corrections.
Minor:
Page 2, 50: what is signal 1 and signal 2 pathway? Please explain in more detail.
3.2 “The effects of Meth…”: the abbreviation “Meth” is not introduced in the text
A table summarizing the main findings and pathway involved for each drug described would be helpful.
Qualit of English Language is good.
Author Response
Reviewer 1
The review manuscript by Guo et al. with a title “Microglia NLRP3 Inflammasome Activation and Neuroimmune Signaling in Substance Use Disorders” is a comprehensive review of the recent literature on drug abuse-induced neuroimflammation. In particular, the review focuses on the signaling pathway of activated microglia in the brain and drugs such as cocaine, methamphetamine, alcohol, opiate-like drugs and marijuana. The review begins by explaining the role of microglia and NLRP3 inflammasome signaling in neuroinflammation. This is followed by several sections, each dedicated to one drug. The review is well and clearly structured with each paragraph having a similar structure. As not many studies have been conducted evaluating the role of abused drugs in neuroinflammation, this interesting review may inspire researchers to conduct further studies in this area of research. The manuscript can be published after some minor corrections.
Responses: Thanks for the reviewer’s positive comments.
Minor:
- Page 2, 50: what is signal 1 and signal 2 pathway? Please explain in more detail.
Response: A figure illustrated signal 1 and signal2 pathway have been added.
- 2 “The effects of Meth…”: the abbreviation “Meth” is not introduced in the text
Response: We added it now.
- A table summarizing the main findings and pathway involved for each drug described would be helpful.
Response: A table has been added.
Reviewer 2 Report
The article is full of inappropriate abbreviations - Mg!!! Written carelessly and very difficult for the reader to perceive.
The article is completely devoid of illustrative material - tables, diagrams, figures. Which are strictly necessary for a review article.
1) I would recommend not to abbreviate the word microglia.
2) Microglia (Mg), the brain resident macrophages, account for 5 - 15% of all brain cells with varying density in different brain regions of human? (ref?). The authors forgot to add the REF.
3) Here, we updated the recent progresses on the effects of abused drugs on NLRP3 inflammasome focusing on brain Mg(?). What does the question mean?
4) Mg, inflammasome, and SUDs. Do the authors think that this title of the chapter is appropriate?
5) 2.1. Updates on Mg biology. The section needs a schema.
6) 2. 2. NLRP3 inflammasome pathway. Schematics are strictly required for signaling pathways!!!!
The text needs additional correction due to the correction of spelling errors and typos.
Author Response
Reviewer 2:
The article is full of inappropriate abbreviations - Mg!!! Written carelessly and very difficult for the reader to perceive. The article is completely devoid of illustrative material - tables, diagrams, figures which are strictly necessary for a review article.
Responses: Thanks for reviewer’s suggestion. We changed Mg into full form and two tables and two figures have been added in appropriate places.
1) I would recommend not to abbreviate the word microglia.
Responses: Now changed.
2) Microglia (Mg), the brain resident macrophages, account for 5 - 15% of all brain cells with varying density in different brain regions of human? (ref?). The authors forgot to add the REF.
Response: Sorry for the omission. More references have been added into the main text.
3) Here, we updated the recent progresses on the effects of abused drugs on NLRP3 inflammasome focusing on brain Mg(?). What does the question mean?
Response: Sorry for the overlook on proofreading. We deleted the “?”.
4) Mg, inflammasome, and SUDs. Do the authors think that this title of the chapter is appropriate?
Response: We have interests on microglia neuroimmune signaling with a focus on NLRP3 inflammasome in the field of substance use disorders. After careful consideration, we believe the title is appropriate but delete the word “Activation”.
5) 2.1. Updates on Mg biology. The section needs a schema.
Response: Yes, a schema has been added.
6) 2. 2. NLRP3 inflammasome pathway. Schematics are strictly required for signaling pathways
Response: Yes, A schematics has been added.
Comments on the Quality of English Language
The text needs additional correction due to the correction of spelling errors and typos.
Response: Yes, we did an additional edit and correction.
Reviewer 3 Report
The review by Guo and colleagues establishes a balance sheet of knowledge between drug abuse and microglial activation/neuroinflammation by focusing on the inflammasome NLRP3. The subject is interesting and topical. Ms. is well structured, clear, and pleasant to read. However, I have a few comments and/or recommendations.
For readers unfamiliar with NLRP3's signaling and activation, the text is sometimes hard to follow. It would have been nice to have illustrations/schema to facilitate understanding of the text and the links between substance use disorders and NLRP3.
I would have appreciated having a paragraph on future therapeutic opportunities, especially since the authors talk about this aspect at the end of their summary and their introduction. What are the potential NLRP3 inhibitors available? Have they been tested on preclinical models of drug dependence? What future research is needed in this area? This type of information is available for other pathologies (see, for instance, doi: 10.1016/j.biopha.2021.111442.), It would have been interesting if the authors discussed this point as well.
Line 33: A reference is missing.
Lines 46–50: The NLRP3 activation process is difficult to follow here since signals 1 and 2 are not yet defined. A schema would also be welcome (see above).
Line 53: A reference is missing.
Line 97: Canonical and non-canonical NLRP3 signaling pathways deserve to be more defined and detailed.
Line 174 and after: Define the abbreviation "Meth."
Author Response
Reviewer 3:
The review by Guo and colleagues establishes a balance sheet of knowledge between drug abuse and microglial activation/neuroinflammation by focusing on the inflammasome NLRP3. The subject is interesting and topical. Ms. is well structured, clear, and pleasant to read. However, I have a few comments and/or recommendations.
Responses: Thanks for the reviewer’s positive comments.
For readers unfamiliar with NLRP3's signaling and activation, the text is sometimes hard to follow. It would have been nice to have illustrations/schema to facilitate understanding of the text and the links between substance use disorders and NLRP3.
Responses: Thanks for the reviewer’s suggestion. A schema has been added now.
I would have appreciated having a paragraph on future therapeutic opportunities, especially since the authors talk about this aspect at the end of their summary and their introduction. What are the potential NLRP3 inhibitors available? Have they been tested on preclinical models of drug dependence? What future research is needed in this area? This type of information is available for other pathologies (see, for instance, [1]. 2021.111442.), It would have been interesting if the authors discussed this point as well.
Responses: Excellent points. A new paragraph has been added to address the questions raised by the reviewer.
Line 33: A reference is missing.
Response: Yes. Reference has been added.
Lines 46–50: The NLRP3 activation process is difficult to follow here since signals 1 and 2 are not yet defined. A schema would also be welcome (see above).
Response: Yes. A schema has been added.
Line 53: A reference is missing.
Response: Reference has been added.
Line 97: Canonical and non-canonical NLRP3 signaling pathways deserve to be more defined and detailed.
Response: Yes. More description and a figure have been added in NLRP3 paragraph.
Line 174 and after: Define the abbreviation "Meth."
Response: Corrected.
Round 2
Reviewer 2 Report
My comments have been taken into account. The article can be accepted for publication